# The Fra-1/AP-1 Oncoprotein: From the “Undruggable” Transcription Factor to Therapeutic Targeting

**DOI:** 10.3390/cancers14061480

**Published:** 2022-03-14

**Authors:** Laura Casalino, Francesco Talotta, Amelia Cimmino, Pasquale Verde

**Affiliations:** 1Institute of Genetics and Biophysics “Adriano Buzzati Traverso”, Consiglio Nazionale dele Ricerche (CNR), 80131 Naples, Italy; amelia.cimmino@igb.cnr.it; 2ReiThera Srl, 00128 Rome, Italy; francesco.talotta@reithera.com

**Keywords:** transcription factor, AP-1 complex, *FOSL1*, Fos-Related-Antigen-1, therapeutic targeting

## Abstract

**Simple Summary:**

Cancer-associated mortality largely depends on metastatic dissemination. Metastasizing cancer cells exhibit drastic phenotypic changes, including the ability to migrate, invade surrounding tissues, survive in the bloodstream, adapt to different microenvironments, and resist to therapeutic treatments. These changes depend on the genetic reprogramming orchestrated by relatively few players (transcription factors). Among the components of the dimeric transcription factor AP-1, the nuclear oncoprotein Fra-1 is strongly implicated in metastasis mechanisms. Therefore, Fra-1, along with other proinvasive transcription factors, represents an ideal therapeutic target. However, as for other DNA-binding proteins, the design of inhibitory drugs is hampered by the structural features of Fra-1. In this Review, after summarizing the Fra-1 functions and mechanisms of accumulation in invasive tumors, we survey the possible application of multiple strategies and emerging technologies aimed at the inhibition of Fra-1 expression and activity, to prevent metastatic dissemination and therapeutic resistance.

**Abstract:**

The genetic and epigenetic changes affecting transcription factors, coactivators, and chromatin modifiers are key determinants of the hallmarks of cancer. The acquired dependence on oncogenic transcriptional regulators, representing a major determinant of cancer cell vulnerability, points to transcription factors as ideal therapeutic targets. However, given the unavailability of catalytic activities or binding pockets for small-molecule inhibitors, transcription factors are generally regarded as undruggable proteins. Among components of the AP-1 complex, the FOS-family transcription factor Fra-1, encoded by *FOSL1*, has emerged as a prominent therapeutic target. Fra-1 is overexpressed in most solid tumors, in response to the BRAF-MAPK, Wnt-beta-catenin, Hippo-YAP, IL-6-Stat3, and other major oncogenic pathways. In vitro functional analyses, validated in onco-mouse models and corroborated by prognostic correlations, show that Fra-1-containing dimers control tumor growth and disease progression. Fra-1 participates in key mechanisms of cancer cell invasion, Epithelial-to-Mesenchymal Transition, and metastatic spreading, by driving the expression of EMT-inducing transcription factors, cytokines, and microRNAs. Here we survey various strategies aimed at inhibiting tumor growth, metastatic dissemination, and drug resistance by interfering with Fra-1 expression, stability, and transcriptional activity. We summarize several tools aimed at the design and tumor-specific delivery of Fra-1/AP-1-specific drugs. Along with RNA-based therapeutics targeting the *FOSL1* gene, its mRNA, or cognate regulatory circRNAs, we will examine the exploitation of blocking peptides, small molecule inhibitors, and innovative Fra-1 protein degraders. We also consider the possible caveats concerning Fra-1 inhibition in specific therapeutic contexts. Finally, we discuss a recent suicide gene therapy-based approach, aimed at selectively killing the Fra-1-overexpressing neoplastic cells.

## 1. Introduction

The oncogenic transcription factors (TFs) were originally envisaged as ideal targets for anticancer therapies [1], and encouraging preclinical/clinical results were obtained by targeting the TFs harboring binding sites for small molecules, such as the Stat3 SH2-phosphorylation domain and the hormone-binding domains of the steroid receptors. Nevertheless, besides the well-characterized DNA-binding domains, most TFs exhibit disordered secondary structures and lack catalytic sites and binding pockets. Thus, differently from therapeutically targeted receptors and cytoplasmic protein kinases, TFs are generally considered “undruggable”.

The AP-1 complex [2,3] results from dimerization between members of the JUN (c-Jun, JunB, and JunD) and FOS (c-Fos, FosB, Fra-1, and Fra-2) families, along with other transcription factors (ATF and Maf families). These proteins share the bZIP domain, in which the DNA-contacting basic amino acid-rich region is flanked by the leucine zipper, which mediates the dimerization, resulting in the large variety of JUN/FOS homo- and hetero-dimers (Figure 1A).

Recently, based on the machine-learning-based AlphaFold method [5], the structures of each component of the human proteome have been predicted and made available [6]. As for the other FOS proteins, the Fra-1 structure can be modeled (with high to very high confidence) only for the 70–80-aa region encompassing the DNA-binding bZIP region, while most of the protein appears intrinsically unstructured (Figure 1B). Therefore, the inherently disordered Fra-1 regions are likely to assume defined structures following the Fra-1/AP-1 interaction with partner molecules.

Several AP-1 components are overexpressed and/or post-translationally modified in response to the major oncogenic pathways. However, various lines of evidence, including the genetic inactivation of individual *JUN* and *FOS* family members in onco-mouse models, show that individual AP-1 proteins can exhibit cell context-dependent oncogenic or tumor suppressor roles, as highlighted in a seminal review entitled: “AP-1: a double-edged sword in tumorigenesis” [4]).

## 2. Structure, Regulation and Functional Roles of *FOSL1*/Fra-1

### 2.1. FOSL1/Fra-1 Structure and Regulation

Among *FOS* family members, the transcription factor Fra-1 is a major driver of cancer cell invasion, EMT (Epithelial-to-Mesenchymal Transition), and metastasis (reviewed in [7,8,9]). The 271 amino acids Fra-1 protein is encoded by the *FOS*-related gene *FOSL1*, localized on chr11q13 [10]. 

*FOSL1* is overexpressed in the aggressive variants of most solid tumors in response to a variety of extranuclear (RTKs, RAS, and BRAF) and nuclear (MYC, AP-1) oncoproteins. Stat3 and Tcf/Lef elements mediate cancer-associated *FOSL1* induction in response to the IL6 and *Wnt*-beta-catenin pathways, respectively (reviewed in [7,8,9]). 

The sequential epigenetic events involved in *FOSL1* transcriptional elongation depend on both upstream and intronic enhancers, controlled by multiple nuclear oncoproteins, such as c-Myc and AP-1. The pathway responsible for the ERK-induced recruitment of c-Myc to the *FOSL1* promoter in response to neuregulin (NRG1) has been recently elucidated in breast cancer [11]. Multiple AP-1 binding sites mediate the *FOSL1* positive autoregulation, which amplifies the effect of Fra-1 posttranslational accumulation. The enhancer-associated epigenetic reader BRD4 drives the recruitment of p-TEFb (positive-Transcription Elongation Factor-b), which phosphorylates the RNAPII (RNA polymerase II) CTD (Carboxy-Terminal-Domain), thus triggering transcriptional elongation by the release of the RNAPII paused on the *FOSL1* promoter [12] (Figure 2A). Notably, the *FOSL1* intronic enhancer is part of a much larger SE (Super Enhancer) region, identified by genome-wide analyses in glioblastoma multiforme (GBM) [13], pancreatic, and colorectal cancer cells [14]. 

*FOSL1* is post-transcriptionally inhibited by multiple miRNAs (Figure 2A). Cancer-associated downregulation of miR-34a/c and miR-15/16-family member miR-497 contributes to the Fra-1-driven neoplastic cell invasion and EMT in breast and colorectal cancer [15,16,17]. Downregulation of miR-19a-3p participates in Fra-1 accumulation in TAMs (Tumor-Associated Macrophages) recruited to breast tumors microenvironment, in which the miR-19a-3p-Fra-1-Stat3 pathway controls the macrophage polarization towards the pro-neoplastic immunosuppressive M2 phenotype [18]. The regulatory mechanisms of miRNA activity include the competition for miRNA binding (sponging) performed by several classes of non-coding RNAs, including the recently characterized circular RNAs (circRNAs). Given their extraordinary stability, circRNAs represent highly effective miRNA sponges [19]. Interestingly, the Genome Browser tracks for circRNAs show the hsa_circ_0022924 [20] deriving from circularization of the *FOSL1* distal exon and including the whole *FOSL1* 3′UTR. Therefore, this cirRNA is a candidate competing for endogenous RNA (ceRNA) sponging the oncosuppressor miRNAs (miR-34 and miR-15/16 family members, along with miR-19a-3p and miR-29a) that downregulate the expression of Fra-1 and other oncoproteins.

The ubiquitin-independent turnover of Fra-1 is prevented by the phosphorylation of S252 and S265 serine residues mediated by ERK2- and Rsk1, respectively [21], and by PKC-theta-dependent phosphorylation of T223 and T230 threonine residues [22] (Figure 2B). In turn, the Fra-1 stabilization in response to the RTK-RAS-RAF-MEK pathway indirectly controls the stability of the c-Jun heterodimeric partner [23].

Accordingly, the BRAF and MEK inhibitors decrease Fra-1 protein accumulation by directly affecting its stability and indirectly abrogating the Fra-1/AP-1-mediated transcriptional autoregulation [24]. In addition, the ubiquitin-independent Fra-1 degradation requires the Fra-1 phosphorylation-independent association with the proteasomal subunit TBP-1 to mediate the proteasomal recognition of the poorly structured Fra-1 C-terminal region [25] (Figure 2B).

### 2.2. Fra-1 in Tumor Growth, Invasion, and Metastasis

Fra-1 overexpression crucially contributes to cancer cell invasion in most solid tumors, including adenocarcinoma (breast, lung, colon, pancreas, and thyroid), squamous cell carcinomas, and non-epithelial cancers, such as melanoma, malignant mesothelioma, and GBM (reviewed in [9]).

Fra-1 drives the morphological changes in cytoskeletal organization, loss of epithelial polarization, increased motility, and invasiveness, which reflect different context-dependent degrees of mesenchymal transformation, from partial to complete EMT [7]. Accordingly, in breast and colorectal adenocarcinoma cell lines the Fra-1-dependent transcriptomes and cistromes comprise well-characterized EMT-inducers, including tyrosine kinase receptors (AXL), EMT-inducing cytokines (TGF-beta and IL-6), EMT-TFs (ZEB1 and ZEB2), and chromatin components (HMGA1) [26,27,28,29,30,31]. In addition to the EMT-related pro-invasive programs, Fra-1 target genes control cell proliferation, survival, and anoikis resistance [32,33,34,35,36,37,38], as summarized in (Figure 2C).

Fra-1 contributes to both autocrine and paracrine mechanisms of EMT and tumor angiogenesis, by inducing multiple cytokines, including TGF-beta in breast and colorectal cancer cells [29,30], and IL-6 and VEGF in the TAMs recruited to tumor microenvironment [18,39,40,41].

Fra-1 downstream effectors also include relevant non-coding transcripts. Fra-1 controls the transcription of the broadly overexpressed onco-miRNA miR-21, which, in turn, contributes to positive feedback loops with AP-1 in RAS-transformed cancer cells [42,43,44]. Another positive feedback is mediated by the Fra-1-dependent control of miR-134 in ovarian cancer. miR-134 inhibits the Protein Phosphatase-1 (PP1) regulatory subunit SDS22, thus potentiating the ERK and JNK MAPK signaling and Fra-1 accumulation and driving cancer cell proliferation, migration, and invasion [45]. Non-coding RNAs also participate to the Fra-1 dependent control of Epithelial to Mesenchymal Transition. For example, the Fra-1-mediated induction of miR-221/222 controls the miR-221/222-TRPS1-ZEB2 pathway, which promotes EMT in breast cancer cells [46]. 

Fra-1 plays a pivotal role in the dynamic balance between cancer and non-cancer stem cells (CSCs). In breast cancer cells, the Twist- and Snail-mediated induction of *FOSL1* results in Fra-1 accumulation, which drives the EMT-associated transition from non-CSCs to CSCs [47]. In colorectal cancer cells, IL-6 potentiates the Fra-1 activity by inducing the HDAC6-mediated Fra-1 deacetylation and accumulation (Figure 2B), resulting in the gain of stem-like features, partially dependent on the Fra-1-mediated transactivation of the *NANOG* promoter [48]. In NF1-mutant GBM tumors and cell lines, *FOSL1* overexpression has been recently implicated in the control of mesenchymal subtype and gain of stem-like features. Accordingly, in a mouse model of GBM, *FOSL1* deletion drives the transition from mesenchymal to proneural transcriptional signature, along with decreased stemness and tumor growth [49]

### 2.3. Fra-1 as Prognostic Biomarker and Cancer Cell Addiction to Fra-1 Overexpression

RNA expression profiling and IHC data show the prognostic relevance of Fra-1 and/or Fra-1-dependent transcriptomes. Time to recurrence and/or metastasis-free survival correlate with Fra-1 expression (alone or in multivariate analyses) in a wide range of adenocarcinomas, including breast [29,34,36,50,51], colon [30,35,41,48], lung [37], pancreas [37], cholangiocarcinoma [52], and squamous cell carcinomas, such as HNSCC (Head and Neck Squamous Cell Carcinoma) [53,54], ESCC (Esophageal Squamous Cell Carcinoma) [55,56], and OSCC (Oral Squamous Cell Carcinoma) [57]), along with non-epithelial cancers, such as glioma [58]. 

Interestingly, in TNBC (Triple-Negative Breast Cancer), the gene signature (Fra-1 classifier) derived from experimentally determined Fra-1-transcriptomes exhibits predictive value superior to most breast cancer prognostic classifiers [34]. Among the therapeutically promising Fra-1-regulated genes in invasive breast cancer [34,36], *ADORA2B* renders the Fra-1-overexpressing TNBCs vulnerable to Adenosine_2b_ receptor inhibitors, such as the common anti-asthmatic theophylline [34]. In multiple tumors, additional synthetic-lethal interactions involve various “druggable” proteins, encoded by Fra-1 target genes and coexpressed with *FOSL1*, including receptors (e.g., *AXL* and *PLAUR*) [28,59,60], cytokines (e.g., *IL6* and *TGFB2*) [26,29,30], and mitotic kinases (e.g., *AURKA*) [37].

The context-dependent roles of Fra-1 expression are pinpointed by the inhibitory effects of *FOSL1* downregulation on tumor growth, detectable in *KRAS*-mutated but not in *KRAS*-wild type PDAC (Pancreatic Ductal AdenoCarcinoma) and LUAD (LUng ADenocarcinoma) cells. As previously shown in RAS-transformed thyroid cells [61], Fra-1 knockdown induces G2-M arrest and apoptosis in *KRAS*-mutated LUAD cells. Accordingly, the knockdown or pharmacological inhibition of Fra-1-controlled mitotic regulators recapitulates the effects of *FOSL1* loss. In *KRAS*-mutated, but not in *KRAS*-wild type lung cancer cells, AURKA depletion selectively blocks cell proliferation and expression of mitotic regulators (*AURKA*, *CCNB1*, *HURP*, *TACC3*, and *PLK1*), though AURKA overexpression is insufficient to rescue all the effects of *FOSL1*-knockdown in *KRAS*-mutated cells [37].

Similarly, *ID1* expression is prognostically relevant in *KRAS*-wild type but not in *KRAS*-mutated LUAD. The ID1 effects on cell proliferation and mitotic machinery largely depend on the ID1-mediated control of *FOSL1*. Interestingly, *FOSL1* re-expression can rescue the *ID1*-silenced phenotype in *KRAS*-mutated cells [62].

Along with *KRAS* mutation, loss of *SMAD4* is a key event in pancreatic cancer progression and metastatic dissemination. Recently, a high-throughput screen for prometastatic SMAD4 target genes has identified *FOSL1*, which is negatively regulated by SMAD4 direct binding to the enhancer region of *FOSL1*. In turn, Fra-1 is necessary and sufficient to recapitulate the effect of SMAD4 loss on metastatic lung colonization [63].

Cancer cell addiction to Fra-1-containing dimers is strongly supported by recent unbiased CRISPR-Cas9 screens to identify dependencies in hundreds of genomically characterized cell lines representing most human cancers [64].

According to the Broad Institute Project Achilles, 205/808 cancer cell lines depend on *FOSL1* expression, while the Sanger’s Cancer Dependency Map shows addiction to *FOSL1* in 50/323 lines (https://score.depmap.sanger.ac.uk, accessed on April 2019). Remarkably, *FOSL1* is unique among *FOS*-family members, which (*FOS* and *FOSB*) are dispensable or (*FOSL2*) essential in only 1/323 lines [65], thus supporting the choice of Fra-1—among FOS proteins—as a target for therapeutic intervention.

### 2.4. Fra-1 in Drug Resistance and Drug Addiction Mechanisms

Together with the unique ability to seed new tumors, CSCs/TICs (Tumor-Initiating Cells) are refractory to anticancer treatments (drug- and radiation-resistant) and so responsible for clinical relapses [66]. The relationship between the EMT-associated transcriptional reprogramming and the gain of stem-like features, including drug resistance [67], is well-established. Therefore, therapeutic targeting of EMT-TFs via Fra-1 inhibition can not only contribute to the eradication of chemo-resistant CSC subpopulations [68], but also antagonize the radiation-resistant CSCs fraction.

BET (Bromodomain and Extra-Terminal domain) inhibitors are currently investigated in several clinical trials addressing hematological malignancies and solid tumors, including breast cancer. The promising therapeutic perspectives of BET inhibitors are hampered by multiple drug-resistance mechanisms, characterized in various preclinical models [69]. The role of Fra-1-containing dimers is suggested by a recent study based on multi-omics profiling and CRISPR functional screening, aimed at identifying the synthetic lethal and resistance interactions with the BET bromodomain inhibitor JQ1 in TNBC. In these cells, Fra-1 regulates its target genes mainly interacting with remote enhancers, which exhibit epigenomic and transcriptional profiles specifically associated with breast cancer subtypes [51,70]. Proteomic analyses by RIME (Rapid Immunoprecipitation Mass spectrometry of Endogenous proteins) show that Fra-1 participates in the BRD4-associated chromatin complexes. In addition, the synthetic-lethal interactions highlight the roles of the *Hippo* and AXL pathways in the resistance to the BRD4 inhibitor [71]. Significantly, Fra-1 cooperates with both the *Hippo* pathway, by interacting with YAP/TAZ/TEAD target promoters [72,73,74]), and the Gas6/AXL pathway, by transcriptionally inducing *AXL* [28,75]). Altogether, these data suggest that Fra-1 therapeutic inhibition might antagonize the acquired resistance to BET inhibitors.

Fra-1 accumulation in melanoma results from the mutationally activated RAS-BRAF-MEK-ERK pathway. Fra-1 triggers a switch in the expression of EMT inducers, involving the ZEB2 and SNAI2 downregulation, associated with the upregulation of ZEB1 and TWIST1, which drive the cancer cell reprogramming leading to melanocyte dedifferentiation and gain of mesenchymal features [76]. As in mammary and breast cancer cells [27,29], the *ZEB1* promoter is regulated by Fra-1, and ZEB1 is a Fra-1 effector in melanoma cells [76]. High levels of ZEB1, correlating with Fra-1 expression and melanoma stemness markers (MITF^lo^/p75^hi^ in CSCs vs MITF^hi^/p75^lo^ in non-CSCs) are implicated in intrinsic resistance to BRAF and MEK inhibitors. In addition, ZEB1 is overexpressed in melanoma cells with acquired drug resistance and in biopsies from patients relapsing while under treatment [77]. Therefore, Fra-1 inhibition might counteract the intrinsic or acquired melanoma resistance to BRAF and/or MEK inhibitors, by suppressing the ZEB1-regulated EMT-like transcriptional programs.

Along with key EMT regulators (*ZEB1* and *AXL*), Fra-1-containing dimers control the transcription of several miRNAs involved in therapeutic resistance. In ovarian cancer, the above-mentioned Fra-1-miR-134 autoregulatory loop causes decreased chemosensitivity to adriamycin and etoposide, because of the miR-134 effect on phosphorylation of the H2AX variant histone, which critically contributes to NHEJ-mediated DNA repair [45].

Although the above-described drug-resistance mechanisms point to Fra-1 inhibition as a tool for restoring the responsiveness to treatments, in specific conditions Fra-1 inhibition might be counterproductive. 

In various neoplastic contexts, acquired resistance to targeted therapeutics depends on the compensating overexpression of some upstream component(s) of the RTK-RAS-BRAF-MEK-ERK signaling pathway. Drug removal results in in vitro growth arrest and in vivo tumor regression, due to the toxic effect of the rebound hyperactivity of the MEK-ERK pathway [78,79]. In melanoma cells exhibiting acquired vemurafenib resistance due to increased BRAF^V600E^ expression, drug removal causes proliferative arrest, which indicates that drug-resistant cells have become addicted to vemurafenib [79]. Accordingly, melanoma patients with acquired resistance exhibit partial therapeutic responses when re-challenged with the same drug after interrupting the treatment [80].

The JunB/Fra-1 heterodimer contributes to the cell death caused by the overdose of MAPK signaling. Following drug removal from dabrafenib- and trametinib-resistant melanoma cells or EGFRi-resistant lung cancer cells, the Mek1/Erk2 rebound activity drives the JunB and Fra-1 accumulation, which triggers proliferative arrest and/or cell death [81]. In several MAPKi-resistant melanoma cell lines harboring different *BRAF* or *NRAS* mutations, the ERK hyperphosphorylation induced by drug withdrawal stimulates the p38-Fra-1-CDKN1A signaling axis, which results in p21 accumulation and proliferative arrest [82]. Moreover, conditioned media from drug-depleted vemurafenib-resistant cells inhibit the growth of untreated cells, thus suggesting the role of some growth-inhibitory secreted factor(s) regulated by Fra-1/JunB [83].

Therefore, in drug-addicted cells subjected to drug withdrawal, Fra-1 inhibition might favor rather than inhibit cancer cell survival. Namely, the clinical benefits resulting from intermittent treatments with RTK, BRAF, or MEK inhibitors could be lost, if *FOSL1* expression is suppressed in coincidence with the proliferative arrest triggered by drug removal.

In addition to conventional and targeted therapies, CSCs are also refractory to immunotherapy, because of the upregulation of immune checkpoint inhibitors such as PD-L1, associated with the presence of M2-polarized macrophages in tumor stroma [84]. Interestingly, Fra-1-containing dimers are involved in both mechanisms. In *KRAS*-transformed human bronchial epithelial cells, Fra-1 contributes to the escape from immune surveillance by mediating the MEK-ERK-dependent induction of PD-L1 [85], while in TAMs, Fra-1 supports the polarization toward the M2 immunosuppressive phenotype [18,26].

### 2.5. Fra-1 in Drug Resistance and DNA Repair Mechanisms

In addition, other Fra-1-regulated mechanisms are implicated in resistance to targeted therapeutics. Based on the synthetic lethality between the loss of PARP activity and BER (Base Excision Repair) defects, PARP inhibitors, such as olaparib, allow the successful treatment of BRCA1/2 mutated cancers, although ineffective in BRCA-wild-type tumors, representing most (80–85%) of TNBCs. Remarkably, PARP1 has been identified among 118 chromatin-bound Fra-1 partners, by proteomic screening in TNBC cells [86]. The interaction between PARP1 and Fra-1 results in reciprocal inhibition. Consequently, while the olaparib-mediated PARP1 inhibition induces Fra-1 expression and activity, Fra-1 (and c-Jun) knockdown sensitizes the TNBC cells to the proapoptotic activity of the PARP inhibitor [87], thus suggesting that Fra-1 therapeutic inhibition could sensitize the BRCA-wild-type TNBCs to treatments with PARP inhibitors.

In the next sections, we will examine several innovative strategies for targeting *FOSL1*/Fra-1 at multiple levels, including the Fra-1/AP-1 DNA-binding activity, *FOSL1* DNA sequence, and mRNA expression, Fra-1 stability, and transactivation mechanisms, along with the recent application of Fra-1-based suicide gene therapies.

## 3. Therapeutic Targeting of *FOSL1*/Fra-1 in Neoplastic Cells

### 3.1. Targeting the Fra-1/AP-1 DNA-Binding Heterodimers

Both polypeptides and small molecule inhibitors have been exploited for interfering with AP-1 activity by several strategies.

The first polypeptide inhibitor of AP-1 was represented by the c-Jun dominant-negative derivative TAM67. This molecule, lacking the N-terminal transactivation domain but retaining an intact DNA-binding domain (Figure 3A), forms homo- and hetero-dimers able to bind to target sequences and suppress the endogenous AP-1 transcriptional activity along with the TPA- or oncogene-induced transformation [88]. The effect of the AP-1 inhibition on neoplastic transformation was corroborated by the in vivo results, showing that TAM67 transgenic expression prevents tumor progression in a skin chemical carcinogenesis system [89].

In the dominant-negative polypeptide A-Fos, derived from the c-Fos bZIP domain, the basic region is replaced with an acidic region, which strongly interacts with the basic region of the JUN-family dimerization partners, to form very stable DNA binding-incompetent heterodimers (Figure 3A). Transfected or adenovirally delivered A-Fos inhibits neoplastic transformation [90] and antagonizes cisplatin resistance in ovarian cancer cells [91]. As for the dominant-negative c-Jun, A-Fos activity has been evaluated by transgenic expression of the dominant-negative c-Fos derivative in a skin carcinogenesis system, in which A-Fos prevents the development of squamous lesions by inducing transdifferentiation into sebaceous tumors [92].

Since in A-Fos the basic residues are replaced with negatively charged amino acids, incompatible with intracellular penetration of the naked protein, the A-Fos delivery requires encapsulation into nanoparticles. This limitation, however, can be circumvented by using smaller c-Jun-interacting peptides displaying stable interhelical hydrophobic interactions. Promisingly, a c-Fos-derived (Fos_169–193_) peptide, containing multiple substitutions in the leucine zipper domain and made cell-permeable by fusion to the arginine-rich HIV-Tat NLS (Tat_48–57_) (Figure 3A), efficiently enters the cell nuclei and inhibits proliferation in breast cancer lines [93]. More recently, the same group has identified a potent and selective Fra-1 inhibitor, by a computational screening of a (>75 million) peptide library, by permutating the sequence encompassing the five leucine zipper heptad repeats. About five hundred in silico-selected (39-aa-long) candidate sequences were further screened in bacteria, by PCA (Protein-fragment Complementation Assay). The selected peptide (Fra1W, Figure 3A) shows a binding affinity for Fra-1 within a nanomolar range and does not homodimerize or heterodimerize with the JUN-family members. Promisingly, the cell-permeable derivative (Fra1W-NLS-Tat) inhibits the AP-1 activity in luciferase reporter assays [94]. 

Furthermore, several small molecules have been characterized as DNA-binding inhibitors by various approaches, including the in silico modeling of the 3D structure of the bZIP domain of the DNA-bound AP-1 complex. From the best-matching decapeptide molecules, non-peptidic small-molecule inhibitors were synthesized by a scaffold-hopping strategy [95].

Among the small-molecule AP-1 inhibitors (reviewed in [96] and represented in Figure 3B), T-5224 has been validated in various preclinical models of non-neoplastic inflammatory and degenerative diseases. The T-5224 therapeutic potential is largely dependent on the inhibition of AP-1-regulated inflammatory cytokines and MMPs (Matrix-degrading MetalloProteases) in models of rheumatoid arthritis [97] and intervertebral disk degeneration [98]. In addition, T-5224 protects from bleomycin-induced lung fibrosis, in which, however, a major role is played by Fra-2/AP-1 dimers in murine alveolar macrophages [99], while the Fra-1/AP-1 dimers exert a protective effect [100] The anti-neoplastic efficacy has been evaluated in a mouse model of HNSCC, in which T-5224 is able to inhibit cancer cell invasiveness and prevent lymph nodal metastases [101].

The bZIP domains and the structure of the DNA-bound complexes are highly conserved among JUN and FOS proteins, thus suggesting that T-5224, modeled on the c-Jun/c-Fos heterodimer, can similarly inhibit the DNA binding by each JUN/FOS heterodimer. Accordingly, T-5224 effectively inhibits AP-1 in TNBC cell lines predominantly expressing Fra-1 [102].

In addition, small molecules can act as AP-1 inhibitors by blocking the TRE sites on DNA. The alkaloid veratramine, identified through virtual screening of a database of natural compounds, selectively binds to the AP-1 target sequence (5′-TGACTCA-3′) by interacting with the DNA minor groove (Figure 3B). Veratramine inhibits the AP-1-DNA interaction and transformed features in EGF-treated mouse keratinocytes, without interfering with MAPK signaling upstream to AP-1. Moreover, despite the structural similarity with the *Smo* (*Smoothened*) inhibitor cyclopamine, veratramine does not affect the Hedgehog pathway [103].

Given the lack of selectivity vs. individual dimers, both polypeptides and small molecules can inhibit the activity of the AP-1 complex not only in cancer cells, in which Fra-1 is a major component, but also in normal tissues, expressing at physiological levels a large variety of AP-1 homo- and hetero-dimers. Notably, JunB, JunD, and c-Fos, can also act as oncosuppressors rather than oncoproteins, depending on the oncogenic lesion and affected cell type [4].

### 3.2. Targeting the FOSL1 Gene and Fra-1 mRNA

The drawbacks of global AP-1 inhibition highlight the importance of developing Fra-1-specific drugs, which should represent better therapeutic bullets.

*FOSL1* expression can be inhibited in cancer cells by several strategies, either irreversibly by gene editing, or reversibly by directly or indirectly targeting the Fra-1 mRNA. Nevertheless, the non-viral tumor-specific delivery represents the bottleneck for most of these strategies. 

Recently, however, therapeutic editing of a breast cancer oncogene (*LCN2*, encoding Lipocalin-2) has been achieved by using an innovative vehicle (tNLG: targeted-NanoLipoGel) for delivery of CRISPR-Cas9 plasmids in TNBC cells. After passing across the leaky tumor endothelial barriers, the deformable tNLG particles are selectively delivered through a cancer cell-specific antibody and conjugated to the surface of the nanoparticles [104] (Figure 4).

The tNLG and similarly designed nanovectors can be utilized for *FOSL1* therapeutic editing in TNBC [26,31,44,47], by exploiting other cell surface molecules, in addition to ICAM-1, for selectively targeting the Fra-1-overexpressing cells. Fra-1 inhibitors could be delivered by using nanoparticles conjugated with antibodies or RNA aptamers [105] binding to the cell surface receptors encoded by Fra-1 target genes (*PLAUR*, *AXL*, and *ADORA2B*). PLAUR being both a target [60] and an upstream regulator [106] of *FOSL1*, is one of the top co-expressed genes in a variety of solid tumors, including breast, lung, colorectal, pancreas, liver, and thyroid adenocarcinoma (with correlation coefficients > 0.5, according to cBioPortal RNA-seq data [107]).

Although most of the currently approved CRISPR/Cas9-based clinical trials are represented by CAR-T cell therapies against hematological malignancies, gene knockout strategies are emerging as powerful antineoplastic tools. The lentiviral delivery of Cas9 and sgRNAs to cancer cells or tumor xenografts has allowed the selective inactivation of the mutated *KRAS* allele [108]. *FOSL1* is activated by overexpression rather than oncogenic mutations. Consequently, given the impossibility to hit a mutated allele, Cas9 and sgRNAs need to be selectively delivered to cancer cells to prevent the possible detrimental consequences of *FOSL1* knockout in non-neoplastic cells.

Differently from *FOSL1* knockout (Figure 4), *FOSL1* knockdown (Figure 5) allows reversible inactivation of Fra-1 expression. 

The therapeutical potential of *FOSL1* silencing has been proven in orthotopic mouse models, in which the shRNA-mediated Fra-1 knockdown suppressed lung metastasis and restored epithelial features of breast cancer cells [34]. Similar results were obtained in other xenograft systems, derived from Fra-1-overexpressing colorectal or pancreatic adenocarcinoma cells [35,37].

In these studies, however, Fra-1 accumulation was suppressed by shRNAs stably expressed in cultured cells before subcutaneous injection in animals. Although the expression of an inducible Fra-1 shRNA showed that regression of established tumor xenografts can be triggered by Fra-1 downregulation [37], Fra-1 silencing by exogenously administered siRNAs has not been tested yet in preclinical models.

Despite the increasing number of clinically approved siRNA-based drugs [109], no oligonucleotide pharmaceutical has been approved for neoplastic diseases. Among the anti-cancer clinical trials registered on ClinicalTrials.gov at the end of 2020 and currently in phase 1/2, 195 are based on ASOs (AntiSense Oligonucleotides), compared to only 17 siRNA-based trials. Inhibition of oncogenic transcription factors is the object of two ongoing trials, aimed at testing the siRNA-mediated inhibition of c-Myc in multiple solid tumors or hepatocellular carcinoma, and seven clinical trials dealing with Stat3-targeted ASOs in advanced cancers [110]. All these drugs are delivered by liposomal carriers. 

An interesting alternative is represented by the recently described siG12D LODER (Local Drug EluteR), for selectively targeting the mRNA encoding the mutated KRAS^G12D^. In the siG12D LODER, the siRNA is incorporated in a biodegradable polymeric matrix, to allow the slow release of the therapeutic oligonucleotide after implantation in pancreatic tumors [111].

While no clinical trial based on *FOSL1* silencing has been registered so far, cancer patients have been experimentally treated with the miR-34a oncosuppressor, which targets the *FOSL1* mRNA. Fra-1 downregulation contributes to the anti-invasive effect of ectopic miR-34a re-expression in colorectal and breast cancer [15,16], thus suggesting the efficacy of miR-34a-based drugs against Fra-1-overexpressing tumors. However, despite promising results in preclinical models, the first-in-human trial of a liposomal miR-34a drug (MRX34) in patients with advanced solid tumors has been terminated before completion of phase I, because of serious adverse events [112]. Current efforts are aimed at chemical modifications and delivery platforms, to prevent the systemic immune activation and maximize the cancer cell-selective delivery of MRX34 and other therapeutic oligonucleotides [110,112].

As an alternative to treatment with MRX34 or similar miRNA mimics, the activity of oncosuppressor miRNAs might be induced by inhibiting their corresponding competing endogenous RNAs (ceRNAs). Given the proposed role of circRNAs as therapeutic targets [19], the knockdown of *FOSL1* circRNA (hsa_circ_0022924), by siRNAs or LNAs targeting the circRNA back splice site, might represent an alternative strategy for rescuing the miRNA-mediated inhibition of Fra-1 expression.

Regarding the challenging delivery of RNA therapeutics, RNA aptamers, which do not require encapsulation in nanovectors and similarly to antibodies facilitate the target-specific delivery, represent a valid alternative to nanoparticles. The recently described AsiCs (Aptamer-linked small-interfering RNA Chimeras) include an RNA-aptamer, mediating the specific binding to the cancer cell surface, joined to a siRNA molecule aimed at silencing the selected mRNA target (Figure 5). Following receptor-mediated internalization, release from the endosomal compartment, and processing by the RNAi machinery, the AsiCs direct the degradation of target transcripts [113]. Treatment with AsiCs mixtures, containing several siRNAs targeting multiple regulators of tumor growth and immune responses, is more efficient than single AsiCs, in mouse models of TNBC. The sequence shared by distinct AsiCs includes the RNA aptamer binding to EpCAM (Epithelial Cell Adhesion Molecule), overexpressed on most epithelial cancer cells [114,115]. In addition to EpCAM, cell surface binding of Fra-1-specific AsiCs might be mediated by aptamers interacting with the above-mentioned cell surface receptors, such as PLAUR, coexpressed with Fra-1.

In addition to gene editing and siRNA-mediated silencing, *FOSL1* can be inhibited at the transcriptional level, by interfering with selected co-activators. Since the same co-activators are also implicated in Fra-1 transactivation mechanisms, both aspects will be discussed in the same section (see below).

### 3.3. Targeting the Fra-1 Protein Stability

Fra-1 can be downregulated at the post-translational level, by promoting protein degradation. Several protein-kinases (ERK1/2, Rsk1, and PKC) induce the Fra-1 accumulation by inhibiting the Fra-1 C-terminal destabilizer region (Figure 2B) and the cancer-associated Fra-1 stability is phosphorylation-dependent. Remarkably, Fra-1 downregulation contributes to the therapeutic effect of the drugs targeting the MAPK pathway [21,22,24]. Therefore, the identification of protein-kinase inhibitors affecting Fra-1 half-life represents an appealing avenue. Alternatively, Fra-1 destabilization could be obtained by recently developed tools (protein degraders), aimed at the forced ubiquitylation and proteasomal degradation of target proteins.

Recently, novel regulators of c-Myc stability in K-Ras-mutated pancreatic cancer cells have been identified by an innovative fluorescence-based screening. The strategy relies on the use of the EGFP-MYC fusion protein as a sensor of c-Myc stability for screening a library of protein-kinase inhibitors (PKIS: Published Kinase Inhibitor Set) [116]. A similarly designed Fra-1-based reporter has been instrumental in investigating the kinetics of ERK activity in living cells [117]. The same construct can be adopted for high-throughput screenings aimed at identifying novel signaling pathways and regulators of Fra-1 phosphorylation and half-life. The reporter cell lines can be generated by stable expression of a reporter construct (FIRE: Fra-1-based Integrative Reporter of ERK), encoding the chimeric protein formed by YFP fused to the 40-aa C-terminal Fra-1 destabilizer, in neoplastic cells exhibiting constitutive MAPK activity driving expression of hyperphosphorylated stable Fra-1 isoforms. Libraries of chemical compounds, such as collections of protein-kinase inhibitors distributed in high-density microwell plates, will be screened by detecting the YFP-associated fluorescence. The candidate Fra-1-destabilizing drugs (positive hits) are revealed as a loss/reduction in yellow fluorescent signal (Figure 6A).

The structurally disordered Fra-1 C-terminal region (Figure 1B) might be directly or indirectly (via-TBP-1 [25]) recognized by the proteasome, and the ERK/Rsk/PKC-mediated phosphorylation might protect the protein from subsequent proteolytic degradation [21,25,118]. The evidence that Fra-1 protein stability was not affected in a non-ubiquitinable mutant protein with all lysines replaced by arginines prompted to suggest that, as for c-Fos, ubiquitylation is at least partially dispensable for Fra-1 catabolism [21]. These results agree with a previous report, showing that overexpression of constitutive MEK1 in human non-neoplastic cells (HEK293) induced both Fra-1 C-terminal phosphorylation and Fra-1 polyubiquitylation, thus suggesting that polyubiquitylation could play roles unrelated to the control of the Fra-1 protein half-life [119].

Recent findings, however, indicate that in several neoplastic cell contexts, polyubiquitylation negatively controls the Fra-1 protein stability. The Ubiquitin-Specific Protease 21 (USP21) is a Fra-1 deubiquitinase contributing to Fra-1 accumulation in *KRAS*-transformed colorectal cancer cells [120]. USP21 knockdown induces Fra-1 polyubiquitylation and decreased stability in the same *KRAS*-transformed CRC cell line (HCT116) in which Fra-1 is stabilized by ERK-mediated phosphorylation [21,120]. Moreover, in a recent study, dealing with a potential anti-tumor agent (xanthohumol) inhibiting ERK activity and Fra-1 accumulation in NSCLC cells, Fra-1 deubiquitylation has been causally linked to the ERK-induced protein stabilization, as shown by the phosphomimetic mutation (S265D) causing the decreased polyubiquitylation and the increased half-life of Fra-1 [121].

Despite the above-summarized discrepancies, possibly reflecting context-dependent roles of Fra-1 polyubiquitylation, we envisage that, at least in part of the Fra-1-overexpressing tumors, degradation of the C-terminally-phosphorylated Fra-1 isoforms can be triggered by forced polyubiquitylation of the protein.

PROTACs (PROteolysis-TArgeted-Chimeras) are small bifunctional molecules acting as bridges between target proteins and well-characterized substrate recognition subunits (CRBN, VHL, and MDM2) for various ubiquitin–E2–E3–ligase complexes [122]. Targeted proteasomal degradation of BRD4 has been obtained by a PROTAC in which a competitive antagonist of the BET bromodomain (JQ1) is joined by a linker to a thalidomide molecule recruiting the cereblon–ubiquitin–ligase complex. Remarkably, the BRD4-destabilizing drug (dBET1), as well as the Stat3 degrader based on a thalidomide-linked molecule of a Stat3 inhibitor, are significantly more effective than parental inhibitors, in murine models of leukemia and lymphoma [123,124], thus highlighting the high therapeutic potential of PROTACs.

Differently from BRD4 and Stat3, for most TFs the identification of ligands is hampered by the lack of binding pockets. Promisingly, however, recruitment of transcription factors by a novel type of degraders (TF-PROTACs), can be mediated by a DNA oligonucleotide containing the TF target site joined by a linker to the VHLL (Von Hippel–Lindau Ligand). Remarkably, the TF-PROTACs containing the NF-kB and E2F consensus sequences trigger the degradation of cognate TFs and inhibit cancer cells proliferation [125]. Given the well-characterized AP-1/TRE interaction, this approach is suitable for the design of AP-1-specific PROTACs (Figure 6B, left section). However, as for the above discussed DNA-binding inhibitors, the AP-1-specific TF-PROTACs will be unable to discriminate between distinct TRE-binding dimers, thus prompting the importance of designing Fra-1-specific PROTACs.

An encouraging option is represented by the peptide-PROTACs (p-PROTAC) strategy (reviewed in [126]), in which a peptide, attached via a linker to the E3 ubiquitin ligase recruiting moiety, mediates the interaction with the target protein. The recently described Fra1W ([94]), a peptidic inhibitor specifically interacting with the Fra-1 leucine zipper, represents a promising candidate as substrate-interacting moiety to develop a Fra-1-specific PROTAC (Figure 6B, middle section). Remarkably, a leucine-zipper-based p-PROTAC has been reported. The degrader molecule, targeting the CREPT (Cell-cycle-Related and Expression-elevated Protein in Tumor) oncoprotein, relies on a 21-aa peptide warhead, which homodimerizes with the target protein through a leucine zipper localized in the C-terminal CREPT CCT (Coiled-Coil-Terminus) domain. In addition to the linker and the VHL ligand, the CREPT-specific PROTAC includes a basic CPP (Cell-Penetrating Peptide), driving the intracellular delivery of the degrader. The antiproliferative effect of this leucine zipper-interacting cell-permeable PROTAC has been validated, both on in vitro cultured pancreatic adenocarcinoma cells, and (by intraperitoneal administration) on the in vivo growth of tumor xenografts [127].

Alternatively, targeted degradation of specific proteins by forced polyubiquitylation could be mediated by intracellularly delivered nanobodies. These are single-domain antibodies consisting of a single monomeric variable domain, with a MW of only 12–15 KDa, and exhibiting antigen-binding specificity comparable to full-size antibodies. The advantages of nanobodies include the possible intracellular delivery. Even if nanobodies are naturally produced by the Camelids immune system, universal synthetic libraries of humanized nanobodies are available [128]. Polypeptidic degraders have been recently obtained by replacing the natural substrate recognition domains of ubiquitin E3 ligases with nanobody miniproteins. To hijack the ubiquitin–proteasome system, the recently described ARMeD (Antibody-Ring-Mediated Destruction) degraders contain a camelid nanobody fused to the RING domain of the ubiquitin E3 ligase RNF4. Given the presence of the RNF4 NLS, these molecules can localize in the nucleus and drive the degradation of nuclear oncoproteins, as shown for PML [129]. 

ARMeD molecules containing Fra-1-specific nanobodies could be exploited for Fra-1 targeted degradation. In addition, similarly to the available Fra-1-S252/S265 phospho-specific antibodies [21], phospho-Fra-1-specific nanobodies will allow generating Fra-1 degraders, selectively targeting the phosphorylated Fra-1 (Figure 6B, right section). These chimeric molecules could efficiently antagonize the Fra-1 accumulation in cancer cells, predominantly overexpressing the C-terminally-phosphorylated Fra-1 isoforms.

The nanobody-mediated interactions, displaying binding affinities in the nanomolar range, offer great advantages in terms of specificity towards target proteins. On the other hand, recombinant ARMeD polypeptides, relatively small to be effectively transferred by electroporation in cultured cells, will require ad hoc systems for clinical applications, such as nanoparticle-mediated delivery of expression vectors to neoplastic tissues. As a proof of concept, the activity of the ARMeD Fra-1 degraders could be tested in TNBC cells, by encapsulating the anti-(phospho)-Fra-1-nanobody-E3-ligase expression vectors in the above-described nanoparticles [104].

### 3.4. Targeting the Fra-1/AP-1-Mediated Transactivation

The first inhibitor of AP-1-mediated transactivation is likely represented by the retinoid molecule SR11302, which is unable to trans-activate the RARE (Retinoic Acid Response Element) but retains intact AP-1 trans-repressing activity [130]. SR11302, however, does not discriminate between distinct JUN/FOS heterodimers, as shown by the SR11302 ability to block both c-Fos/AP-1 heterodimers in retinal angiogenesis [131], and Fra-1/AP-1 heterodimers in HNSCC metastatic progression [53,101].

More recently, pharmacological modulation of transcriptional coregulators has become a promising area of therapeutic intervention [132], as well shown in the case of BRD4 inhibitors. JQ1 is a competitive inhibitor of the interaction between the bromodomains of BET-family epigenetic readers (BRD2, BRD3, and BRD4) and acetylated histones [133]. A high abundance of chromatin-associated BRD4 (and MED1) is a distinctive feature of SEs [14]. Because of the very high levels of chromatin-associated BRD4, SEs (but not standard enhancers) are inhibited by JQ1 [13,69]. JQ1 and related drugs affect a relatively small number of oncogenes, including *MYC* and *FOSL1*. While *MYC* is the key target of JQ1 in hematopoietic cancers, *FOSL1* is a major JQ1 target in lung adenocarcinoma [134] and other solid tumors, such as osteosarcoma and childhood sarcoma, in which Fra-1 downregulation phenocopies the effect of the BRD4 inhibitor [135,136]. 

In addition to being regulated by a BRD4-dependent SE, Fra-1/AP-1 transactivates several target genes controlled by SEs. In HNSCC, Fra-1/AP-1 controls key pro-metastatic genes (such as *FOSL1* itself, *SNAI2*, and miR-21) through super-enhancers, which recruit the coactivators (MED1 and BRD4) through Fra-1-dependent mechanisms [53,137].

Along with BET inhibitors, novel drugs, blocking the Fra-1 interaction with chromatin-bound coactivators, will allow interfering with the Fra-1-mediated transactivation (Figure 7). The action of these drugs will be amplified by the positive autoregulatory effect of Fra-1/AP-1 on the *FOSL1* gene.

In a large-scale screening for combinatorial interactions between transcription factors, the bromodomain-containing epigenetic reader TRIM24 was identified among 23 Fra-1 interactors [138]. TRIM24, which contains both a PHD- and a bromodomain interacting with the dual histone mark H3K4me0/H3K23Ac, is an oncogenic transcriptional coactivator overexpressed in breast [139] and prostate [140] adenocarcinoma, along with glioblastoma, in which TRIM24 is a Stat3 coactivator [141]. The Fra-1-TRIM24 interaction, pointing to TRIM24 as a Fra-1 coactivator, suggests that Fra-1 transcriptional activity can be hampered by TRIM24 inhibitors. While the first-generation TRIM24-specific drugs exerted poor antiproliferative effects [142], a selective TRIM24 degrader (dTRIM24), generated by the PROTAC strategy, inhibits cell proliferation and survival in acute leukemia [143]. It will be interesting to investigate the dTRIM24 antineoplastic activity in solid tumors, along with the possible role of the inhibition of Fra-1-mediated transactivation.

More recently, screening for proteins interacting with the chromatin-bound Fra-1, the RNA helicase p68/DDX5 has been characterized as an oncogenic coactivator of Fra-1 in TNBC [86]. DDX5 is directly bound and destabilized by resveratrol, a well-known chemopreventive nutraceutical. DDX5 downregulation contributes to the antiproliferative effect of resveratrol in prostate cancer cells [144]. Since AP-1 is a major target of the resveratrol chemopreventive activity [145], DDX5 inhibition might contribute to the tumor suppression by resveratrol in Fra-1-overexpressing cancers.

### 3.5. Fra-1-Based Suicide Gene Therapy Strategies

In addition to the approaches based on the inhibition of Fra-1 expression or activity, other therapeutic strategies are based on the Fra-1 stabilization mechanisms in neoplastic cells. Recently, a suicide gene therapy approach has been proposed, based on the phosphorylation-dependent Fra-1 stability control mechanism (Figure 8).

To drive the selective killing of cancer cells overexpressing the C-terminally-phosphorylated Fra-1 isoforms, a fusion protein (NLS-HSVtk-Fra-1-163-271) has been generated by replacing the YFP reporter with the HSV-tk selectable marker in the above-mentioned reporter construct (FIRE) [117]. The herpes virus thymidine kinase converts the prodrug (ganciclovir) in a cytotoxic nucleotide precursor, that causes double-stranded breaks when integrated in DNA. In this construct, the Fra-1 domain containing the phosphorylation-dependent destabilizer is fused to the C-terminus of HSVtk. The resulting HSVtk-Fra-1 fusion protein accumulates exclusively in cancer cells overexpressing Fra-1-phosphorylating activity, making these cell populations vulnerable to ganciclovir treatment, while normal cells are spared (Figure 8). Promisingly, orthotopically propagated GBM xenografts can be growth-inhibited in vivo by intracranial administration of retroviruses expressing the HSVtk-Fra-1 suicide gene [146].

## 4. Concluding Remarks

*FOSL1*, along with the cognate transcript and protein product, represents a highly promising therapeutic target in a wide range of aggressive tumors, in which Fra-1-controls the key hallmarks of tumor progression, including cancer cell proliferation, EMT, gain of stem-like features, anoikis resistance, and metastasis.

In agreement with the title of a recent review (“Targeting transcription factors in cancer-from undruggable to reality” [147]), here we have outlined several strategies aimed at the design and delivery of *FOSL1*/Fra-1/AP-1-targeting molecules. We have previously referred to Fra-1 as “a transcription factor knocking on therapeutic applications’ door” [8]. The increasingly large arsenal of therapeutic weapons, from tumor-specific nanoparticles to proteasome-targeting chimeras, along with Fra-1-based suicide gene therapies, indicates that the time has definitely come, to unlock the Fra-1 “therapeutic applications’ door” [8].

## Figures and Tables

**Figure 1 cancers-14-01480-f001:**
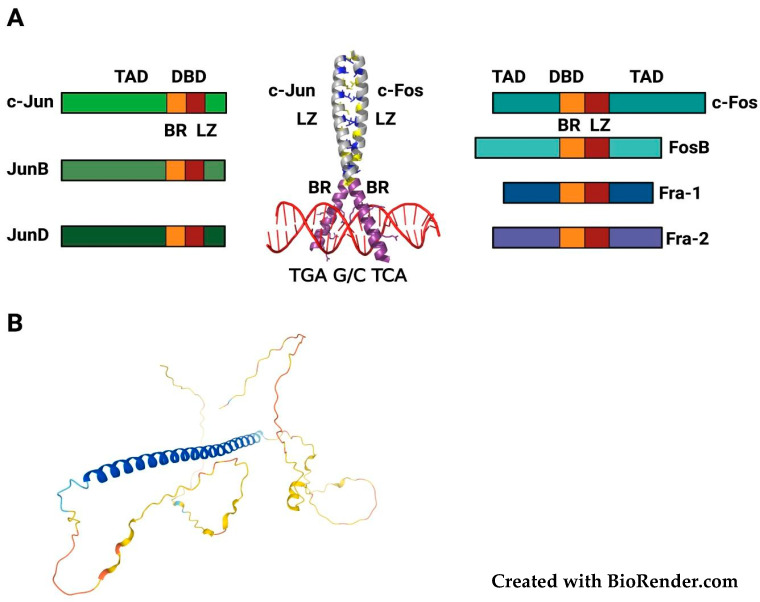
(**A**) The JUN/FOS AP-1 complex (modified from [4]): **Left**: the JUN proteins (c-Jun, JunB, JunD), which form both homodimers with other JUN-family members and heterodimers with FOS proteins. **Right**: the FOS-family members (c-Fos, FosB, Fra-1, Fra-2), which can only heterodimerize with JUN proteins. BR: Basic Region. LZ: Leucine Zipper. TAD: Trans-Activation Domain. **Middle**: the crystal structure of c-Fos/c-Jun DNA-bound bZIP domain (PDB: 1FOS). The X-shaped alpha-helical structure contains the LZ (gray) and the BR (purple), directly interacting with the DNA (red). LZ leucines (blue) and other hydrophobic residues (yellow) are indicated and the canonical heptameric TRE (TPA-Response Element) sequence is reported (5′-TGA G/C TCA-3′). (**B**) 3D structure of human Fra-1. The model is based on the AlphaFold prediction method (available at the AlphaFold Protein Structure Database, https://alphafold.ebi.ac.uk/entry/P15407, accessed on 1 July 2021). The N-terminus of the protein is on the upper left. The colors represent the per-residue model confidence score. Blue: very high. Light blue: confident. Yellow: low. Orange: very low. The blue-colored predicted alpha-helix (from Ser101 to Pro175) encompasses the bZIP DNA-binding domain, while the other regions are likely unstructured, in absence of binding to Fra-1 interaction partners.

**Figure 2 cancers-14-01480-f002:**
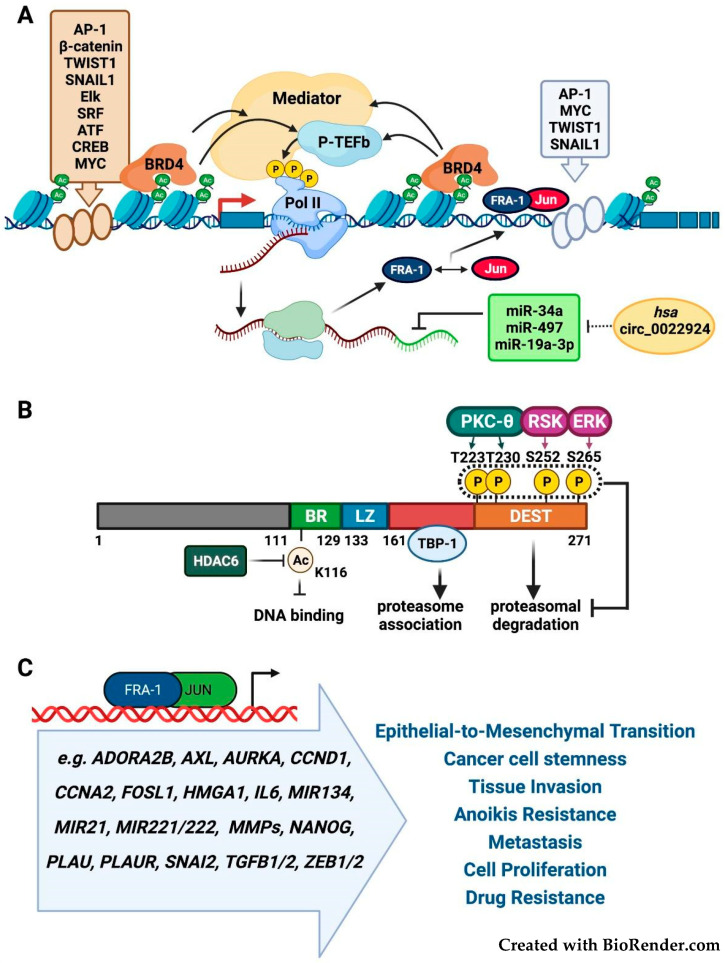
(**A**) *FOSL1* transcriptional and post-transcriptional regulation. The major oncogenic pathways control the *FOSL1* transcription through multiple TFs interacting with the *FOSL1* promoter region (AP-1, beta-catenin, TWIST1, SNAIL1, Elk, SRF, ATF, and CREB) or the intronic enhancer (AP-1, MYC, TWIST1, and SNAIL1). The BRD4 association with acetylated histones drives the recruitment of P-TEFb, which triggers transcriptional elongation by phosphorylating the CTD of RNPII stalling on the *FOSL1* promoter. The *FOSL1* positive autoregulation is represented by the Fra-1-containing heterodimers binding to AP-1 sites in the intronic enhancer region. miR-34a, miR-497, and miR-19a-3p (green box) represent the oncosuppressor miRNAs targeting the Fra-1 mRNA 3′UTR. Hsa_circ_0022924 indicates the circRNA encompassing the *FOSL1* distal exon, containing the 3′UTR. The dashed line represents the possible sponging of the miRNAs by Hsa_circ_0022924. (**B**) Fra-1 post-translational modifications. The diagram represents the major Fra-1 domains, including the basic region (BR), leucine zipper (LZ) and the C-terminal destabilizer region (DEST) implicated in Fra-1 proteasomal degradation. The ERK- and RSK- phosphorylated (S252 and S265, respectively) and the PKC-theta-phosphorylated (T223 and T230) residues, protecting Fra-1 from degradation, are shown along with the (TBP-1-binding) region involved in proteasome association. The acetylated residue (K116), localized in the BR and subjected to HDAC6-mediated deacetylation, negatively controls the Fra-1/AP-1 DNA-binding activity (modified from [8]). (**C**) The major functionally characterized target genes and neoplastic hallmarks controlled by Fra-1 in tumor progression.

**Figure 3 cancers-14-01480-f003:**
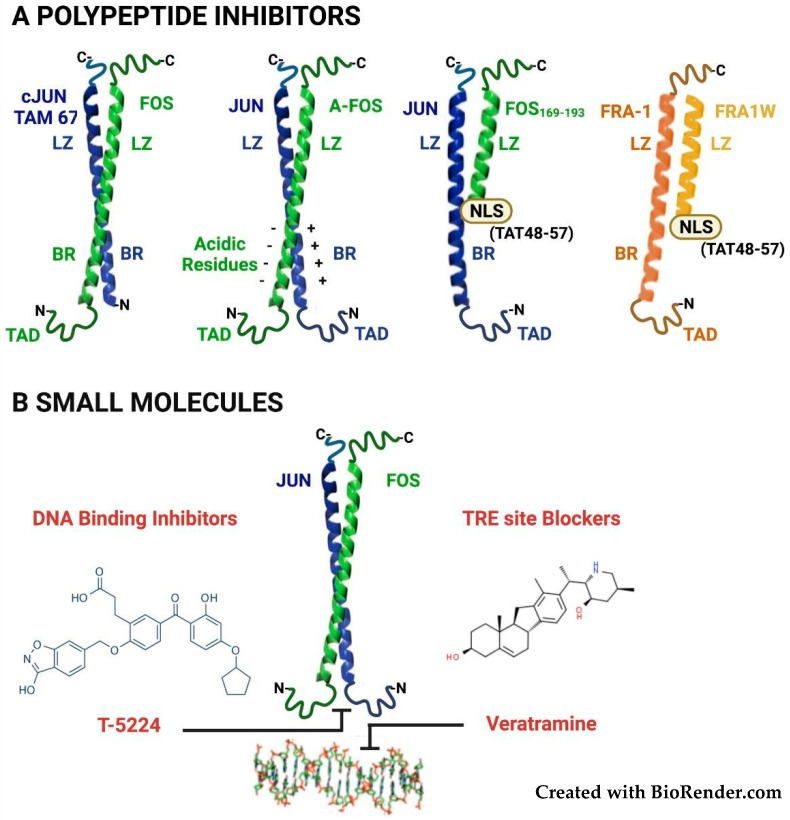
Inhibition of Fra-1/AP-1 DNA binding activity. (**A**) Polypeptide inhibitors. The recombinant or synthetic polypeptides act as bZIP competitors. The JUN/FOS dimers are represented as X-shaped coiled-coil structures, with the N- and C-terminal regions (not in scale) flanking the central DNA-binding domain. The leucine zipper (LZ), basic region (BR) and the transactivation domain (TAD) domains are indicated. Blue: JUN proteins, green: FOS proteins, Orange: Fra-1. TAM67 is a deletion mutant of c-Jun lacking the N-terminal TAD. TAM67 acts as a dominant-negative on AP-1-dependent transcription by sequestering the AP-1 proteins in inactive homo- and hetero-dimers. The 68-aa dominant-negative A-Fos polypeptide, generated by replacing the BR of the c-Fos bZIP with an amphipathic 25-aa acidic extension forms very stable heterodimers, which subtract from DNA-binding the JUN-family partners. The Fos169-193/Tat48-57 polypeptide, encompassing a 25-aa segment derived from the c-Fos LZ forms DNA binding-incompetent heterodimers with the JUN-family partners. The peptide is rendered cell-permeable by fusion to the 10-aa Tat-derived Nuclear Localization Signal (NLS). The 39-aa Fra1W (yellow), derived from the Fra-1 LZ, forming very stable Fra-1-Fra1W homodimers, specifically inhibits the Fra-1/AP-1 activity by sequestering Fra-1 from its heterodimeric partners (the Fra1W-NLS-Tat represents the cell-permeable derivative). (**B**) Small-molecule AP-1 inhibitors. The small-molecule DNA-binding inhibitor T-5224 blocks the AP-1 binding by interacting with the basic region of the JUN/FOS heterodimers. The small-molecule inhibitor veratramine specifically interacts with the TRE site, thus preventing the AP-1 binding to DNA.

**Figure 4 cancers-14-01480-f004:**
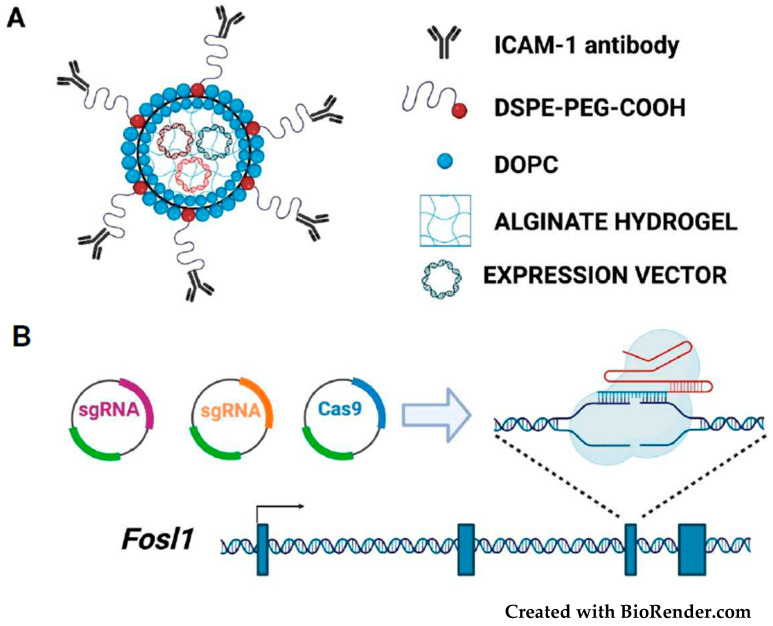
*FOSL1* editing in TNBC cells: (**A**) The multilayer structure of the tumor-targeted nanolipogel (tNLG) (modified from [104]). The plasmid-loaded hydrogel is coated with a lipid bilayer formed by the zwitterionic DOPC and the anionic DSPE-PEG. The hydrophilic tail of the DSPE-PEG mediates the binding of the ICAM-1 antibody, which allows for the specific delivery of plasmid vectors (or siRNAs) to TNBC cells. (**B**) The gRNAs and Cas9 expression vectors are encapsulated in the tNLG nanoparticle. The diagram shows the *FOSL1* editing by a representative gRNA-Cas9 ribonucleoprotein complex targeting the third exon, encoding the essential bZIP region.

**Figure 5 cancers-14-01480-f005:**
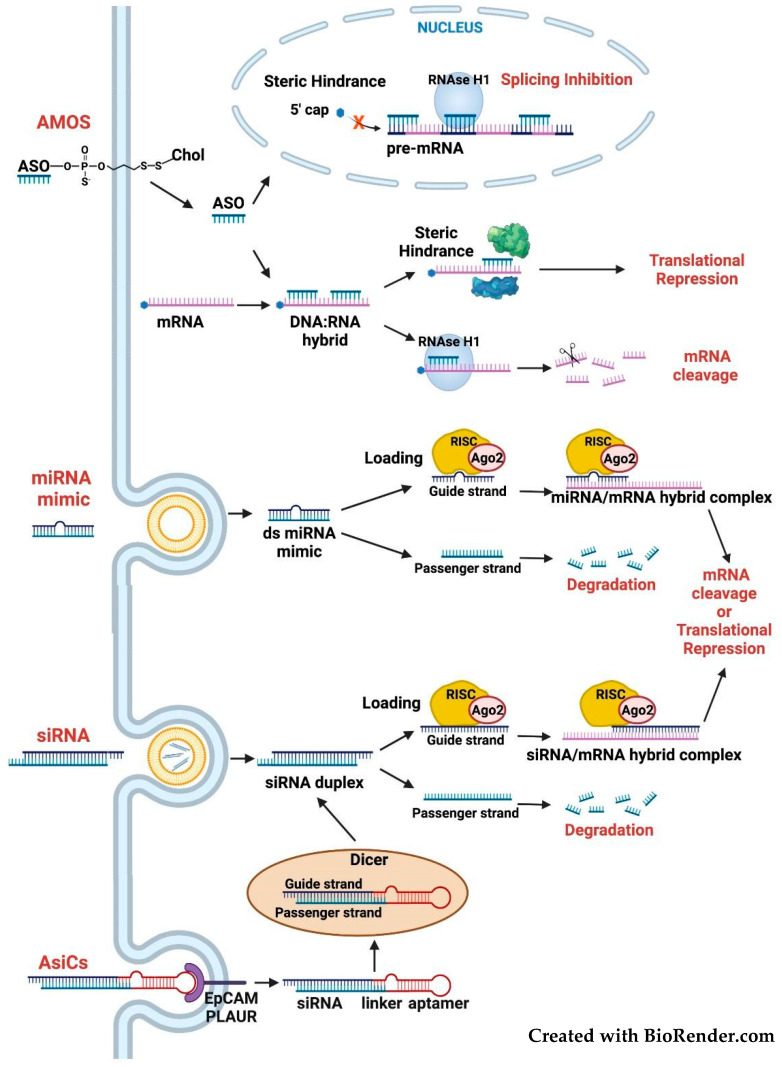
Inhibition of *FOSL1* expression. Different inhibitory mechanisms of distinct oligonucleotides (AMOs, miRNA mimics, siRNAs, and AsiCs) targeting the Fra-1 mRNA. AMOs: depending on the complementary mRNA region and oligonucleotide chemical modifications, the AMOs (Antisense Modified Oligonucleotides) can function by steric hindrance, to block the pre-mRNA splicing or mature mRNA translation, or by triggering the RNAseH-mediated degradation of the DNA–RNA (or gapmer–RNA) hybrid formed with the target transcript. siRNAs and miRNAs: both oligonucleotides are incorporated in the cytoplasmic RISC complex. While the siRNAs trigger the endonucleolytic cleavage of the fully complementary Fra-1 transcript, the partially complementary miRNAs induce translational repression and degradation of the Fra-1 mRNA (and other co-targeted transcripts). While the chemically modified AMOs (amphipatic, with hydrophobic groups exposed) exhibit good tissue distribution, the poorly distributed (hydrophilic) double-stranded siRNAs and miRNA mimics require encapsulation in liposomal nanoparticles, as in the case of the miR-34a-derived drug (MRX34). AsiCs (Aptamer-linked small-interfering RNA Chimeras): these oligonucleotides do not require encapsulation in liposomal carriers, since the internalization is mediated by the binding of the aptamer portion to the cell surface receptor. The release of siRNAs from the internalized AsiC molecules is mediated by Dicer.

**Figure 6 cancers-14-01480-f006:**
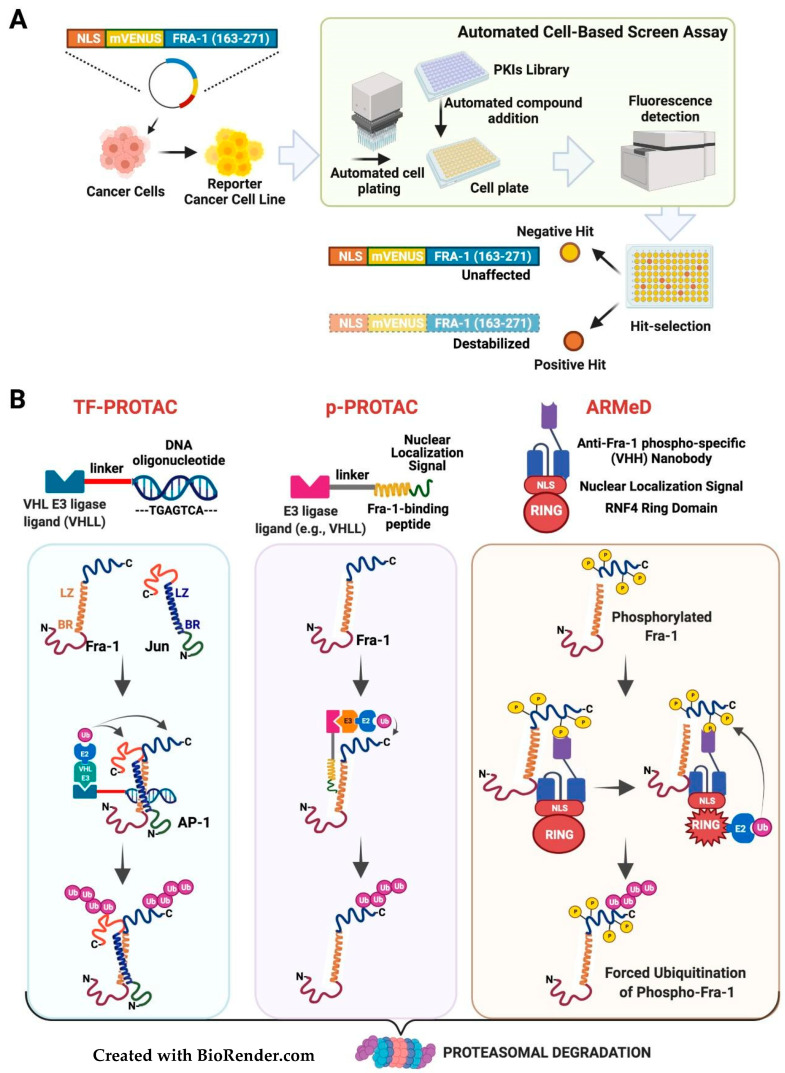
Inhibition of Fra-1 protein stability: (**A**) Identification of Fra-1 destabilizers by fluorescence-based high-throughput screening of chemical libraries. A cancer cell line expressing the hyperphosphorylated Fra-1 isoforms is stably transfected with the reporter construct (FIRE = Fra-1-based Integrative Reporter of ERK), expressing the YFP fused to the Fra-1 C-terminal region. By automated systems, libraries of protein kinase chemical inhibitors (e.g., the PKIS, Published Kinase Inhibitor Set) are screened by cell-based assays based on readout detection of the YFP-associated fluorescence. The candidate Fra-1-destabilizing drugs (positive hits) are revealed as a loss/reduction in yellow fluorescent signal. (**B**) Design of Fra-1 degraders. Three different molecules aimed at forcing the Fra-1 polyubiquitylation and proteasomal degradation are represented. **Left** panel: TF-PROTAC. The small-molecule ligand (VHLL) of the E3 ubiquitin ligase VHL is connected by a linker to a DNA oligonucleotide encompassing the AP-1 binding sequence (TRE). After binding to the oligonucleotide warhead, the AP-1 heterodimers are subjected to polyubiquitylation by the E2 ubiquitin-conjugating enzyme recruited by the E3 ligase bound to the cognate ligand (VHLL). Since the TRE-binding heterodimers can include each JUN and FOS family member, this strategy does not allow the selective targeting of Fra-1. **Middle** panel: p-PROTAC (peptide-PROTAC). The small-molecule ligand of an E3 ubiquitin ligase (e.g., VHLL) is connected by a linker to the Fra-1-binding peptide, represented by the Fra1W, which specifically interacts with the Fra-1 leucine zipper (Figure 3A). The LZ-bound PROTAC recruits Fra-1 to the E3 ligase, and the E2 ubiquitin-conjugating enzyme catalyzes the Fra-1 polyubiquitylation. While the ubiquitin-tagged Fra-1 undergoes proteasomal degradation, the p-PROTAC molecule is recycled. NLS: the HIV-Tat-derived basic Cell-Penetrating Peptide appended to the Fra1W (Figure 3A), to allow the entry and nuclear localization in target cells. **Right** panel: ARMeD (Antibody-Ring-Mediated Destruction). The Fra-1 degrader contains the 15KDa nanobody fused to a portion of the ubiquitin E3 ligase RNF4. Following the nanobody-mediated binding to substrate and RING activation, the chimeric molecule, containing the RNF4 NLS and RING domain recruits the E2 ubiquitin-conjugating enzyme, which catalyzes the Fra-1 polyubiquitylation and proteasomal degradation. The phospho-S252/S265-specific ant-Fra-1 nanobody is aimed at selectively targeting the highly phosphorylated Fra-1 isoforms accumulated in cancer cells. In vivo delivery of the ARMeD polypeptides can be mediated by expression vectors encapsulated in ad hoc nanoparticles, as in Figure 4.

**Figure 7 cancers-14-01480-f007:**
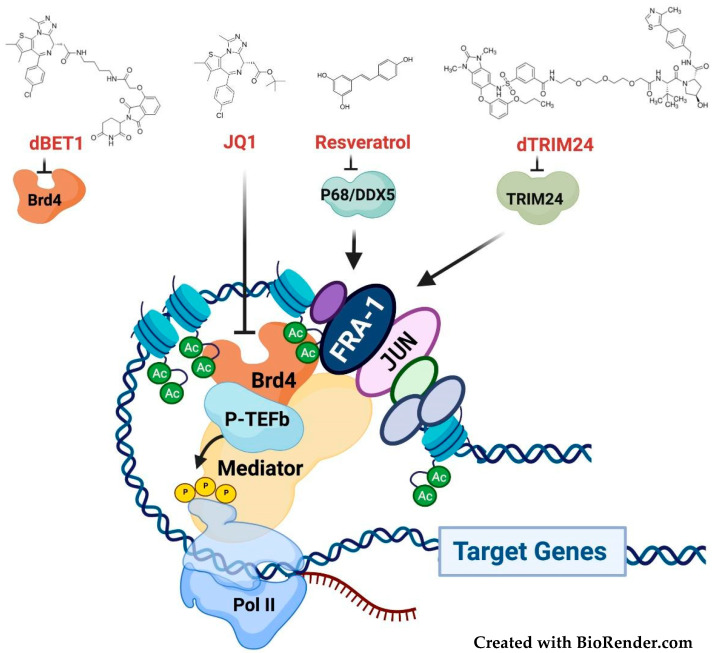
Inhibition of the Fra-1-mediated transactivation. The cartoon shows a representative enhancer-bound Fra-1/AP-1 complex, along with other TFs and transcriptional coactivators. Fra-1-regulated enhancers, including the *FOSL1* enhancer, are often part of highly BRD4-dependent super-enhancers, driving the transcription of key oncogenes. Along with BRD4, the chromatin-bound cofactors TRIM24 and p68/DDX5, identified by PPI (Protein-Protein Interaction) screens, are implicated in Fra-1-mediated transactivation. Both BRD4 and TRIM24 can be targeted by the available PROTAC degraders (dTRIM24 and dBET1, respectively). The dBET1 substrate-binding moiety is the JQ1 inhibitor, which prevents the bromodomain-mediated interaction of BRD4 with histones acetyl-lysine residues, as shown. Given the *FOSL1* positive autoregulation, the same inhibitors can affect the Fra-1 transcriptome both directly, by inhibiting the Fra-1 activity, and indirectly, by decreasing the *FOSL1* expression and Fra-1 accumulation.

**Figure 8 cancers-14-01480-f008:**
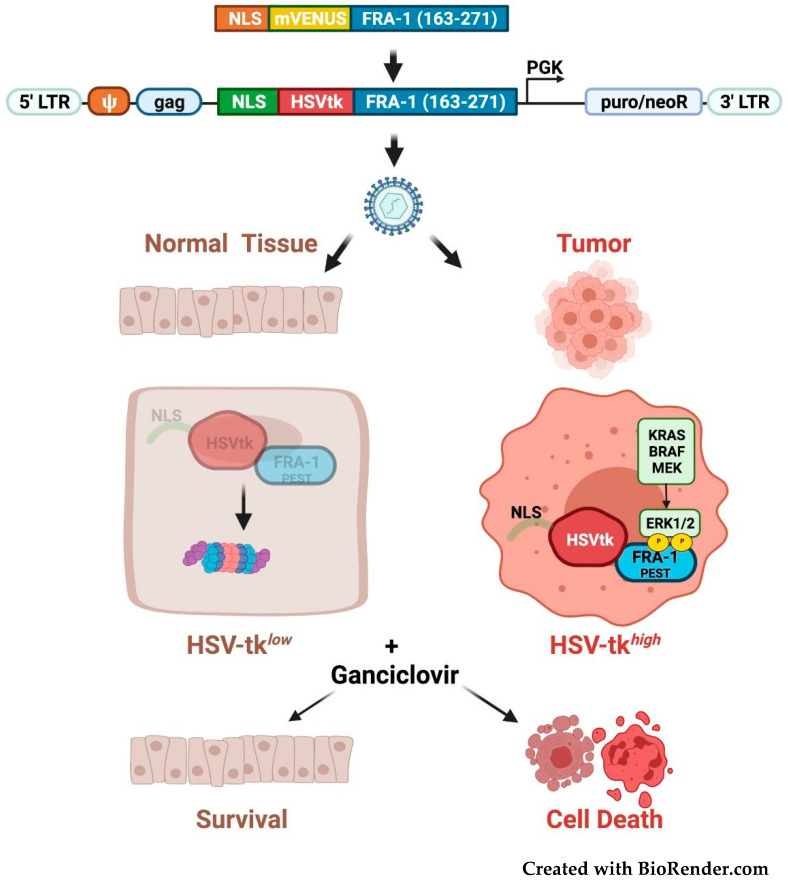
Suicide gene strategy for targeting cancer cells expressing constitutively stable Fra-1 isoforms. Structure of the lentiviral expression vector containing the HSV thymidine kinase gene fused to the Fra-1 C-terminal region (aa. 163–271) encompassing the ERK-dependent destabilizer. NLS: Nuclear Localization Signal. The lentiviral particles infect both normal and cancer cells, as shown. While in normal cells the Fra-1 destabilizer drives the chimeric protein to proteasomal degradation, the cancer-associated constitutively active MEK-ERK pathway induces Fra-1 S252/S265 phosphorylation. Consequently, the tk-Fra-1 fusion protein is protected from proteasomal degradation and accumulation. Following ganciclovir treatment, the thymidine kinase-mediated transformation of the prodrug causes the accumulation of toxic metabolites blocking DNA replication and triggering cell death in phospho-Fra-1-expressing cancer cells, while normal cells are spared.

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
