# Peer review of "The Fra-1/AP-1 Oncoprotein: From the “Undruggable” Transcription Factor to Therapeutic Targeting"

_cancers, 2022, doi:10.3390/cancers14061480_

Round 1
Reviewer 1 Report
In their paper Casalino et al review the literature on AP1 family member Fos-Related Antigen-1 (FRA1) with the focus on the therapeutic inhibition of its function. The authors of this review have been working in the area for many years, their contribution to the field is very important.
The review summarizes the most important studies on function and regulation of FRA-1 in human cancer. FRA-1 as well as other transcription factors are poorly druggable targets. However, the authors analyse a number of direct and indirect approaches, which (although with different degrees of specificity) allow to interfere with FRA-1 functions in therapeutic settings.
This is a very well-written comprehensive paper, timely and highly relevant to cancer. It will attract the attention of the broad community of cancer researchers interested in signal transduction and drug development.
Minor points
- Figure legends need to be restructured and individual panels separately explained. For example, in line 56 “FOSL1 transcriptional and posttranscriptional regulation” belongs to Figs 1b and 1c; this needs to be indicated. Layout in Figures 2 and 3 is of poor quality.
- Figure 2. It would be useful to show all c-FOS-derived peptides in the same colour.
- Line 58. “…multiple TFs interacting with the FOSL1 5’UTR…” FOSL1 promoter?
- A very recent study has shown that c-MYC binds and activates FOSL1 gene promoter (Le Shu et al., 2022).
- Ser252 and Ser265 are phosphorylated by RSK and ERK respectively.
- FRA-1 function in DNA repair (interaction with PARP, impact on NHEJ pathway) is discussed in different parts of the section 2.4. These findings seem to be interconnected and can be presented and discussed in a separate paragraph.
- It has been shown in various cell lines that the bulk of FRA1 protein is degraded independent of ubiquitylation. However, a fraction of the protein can be ubiquitylated, which may affect FRA1 function, without causing protein degradation (Hoffmann et al., 2005). Is artificially ubiquitylated phosphorylated FRA1 unstable? This needs to be discussed in the section 3.3., in the context of PROTAC.
- Line 529. … between target proteins and well-characterized receptors… between target proteins and substrate recognition subunits…?
- Line 616. RARE, retinoid acid response element?
Author Response
In their paper Casalino et al review the literature on AP1 family member Fos-Related Antigen-1 (FRA1) with the focus on the therapeutic inhibition of its function. The authors of this review have been working in the area for many years, their contribution to the field is very important.
The review summarizes the most important studies on function and regulation of FRA-1 in human cancer. FRA-1 as well as other transcription factors are poorly druggable targets. However, the authors analyse a number of direct and indirect approaches, which (although with different degrees of specificity) allow to interfere with FRA-1 functions in therapeutic settings.
This is a very well-written comprehensive paper, timely and highly relevant to cancer. It will attract the attention of the broad community of cancer researchers interested in signal transduction and drug development.
Minor points
- Figure legends need to be restructured and individual panels separately explained. For example, in line 56 “FOSL1 transcriptional and posttranscriptional regulation” belongs to Figs 1b and 1c; this needs to be indicated. Layout in Figures 2 and 3 is of poor quality.
In response to these precious suggestions, we have now:
- extensively modified the Figure Legends.
- Subdivided and reorganized several Figures (eight figures in the revised version).
- Changed the layout of Figures 2 and 3.
- Figure 2. It would be useful to show all c-FOS-derived peptides in the same colour.
We have now represented all c-FOS-derived peptides in the same colour (green), In addition, in response to Reviewer #3, raising our attention on the c-Jun-derived dominant-negative TAM67, we have added to the drawing the unstructured regions flanking the bZIP domain on both N-terminal and C-terminal sides.
- Line 58. “…multiple TFs interacting with the FOSL1 5’UTR…” FOSL1 promoter?
Corrected to FOSL1 promoter region
- A very recent study has shown that c-MYC binds and activates FOSL1 gene promoter (Le Shu et al., 2022).
Thank you for the suggestion on the recent study on the NRG1-Myc-Fra-1 pathway, that we had missed. We have now referred to this novel regulatory axis and introduced the pertinent reference (lines 98-100).
- Ser252 and Ser265 are phosphorylated by RSK and ERK respectively.
Now properly specified in the revised text (lines 145-146).
- FRA-1 function in DNA repair (interaction with PARP, impact on NHEJ pathway) is discussed in different parts of the section 2.4. These findings seem to be interconnected and can be presented and discussed in a separate paragraph.
We have now discussed this important aspect, in a novel paragraph (2.4., entitled: Fra-1 in drug resistance and DNA repair mechanisms, lines 322-337) in the revised text.
- It has been shown in various cell lines that the bulk of FRA1 protein is degraded independent of ubiquitylation. However, a fraction of the protein can be ubiquitylated, which may affect FRA1 function, without causing protein degradation (Hoffmann et al., 2005). Is artificially ubiquitylated phosphorylated FRA1 unstable? This needs to be discussed in the section 3.3., in the context of PROTAC.
We thank the reviewer for raising our attention on the contrasting evidence on the role of polyubiquitylation in the control of Fra-1 half-life. We have now expanded the discussion and included the citation (ref 121) of a recent article dealing with mechanisms of ERK-dependent Fra-1 stabilization in NSCLC (lines 571-619).
- Line 529. … between target proteins and well-characterized receptors… between target proteins and substrate recognition subunits…?
Corrected.
- Line 616. RARE, retinoid acid response element?
Specified in parentheses and added to the List of Abbreviations.
Reviewer 2 Report
The review paper “The Fra-1/AP-1 oncoprotein: from the “undruggable” transcription factor to therapeutic targeting” by Casalino et al. describes functional roles of Fra-1 in tumor growth, invasion, metastasis, prognosis and drug resistance. Moreover, the paper summarizes strategies targeting Fra-1 expression, stability, and transcriptional activity, as well as tumor-specific delivery of Fra-1/AP-1-specific drugs. AP-1 transcription factors, including Fra-1, are generally considered “undruggable”. Therefore, the review is interesting and highlights the potential to develop novel agents targeting transcription factors for future cancer treatment.
However, there are several problems that reduce the overall quality of this paper. Here are some suggestions for improvement.
- The quality of the figures is poor and must be improved.
- Fig.1A middle is the same as Figure1 in“AP-1: a double-82 edged sword in tumorigenesis” (https://www.nature.com/articles/nrc1209). However, there is no citation in the figure legend.
- The figure legends are unclear and need rewording.
- There is no indication of panels A-D and A-B in the legend of Figure 1 and Figure 3, respectively.
- The Figures 2, 3 and 5 are difficult to read. Please change the font.
- Sections 2.2, 2.3 and 2.4: graphical display or/and a table to summarize the functional roles and the underlying mechanisms of Fra-1 in tumors will help to make the message clearer.
- Lines 157-164: the sentences could be phrased better.
- Lines 318-326: The description of Omomyc can be omitted, since it’s not relevant to Fra-1/AP-1.
- Sections 3.2 and 3.3: The discussion of the strategies targeting Fra-1, which are under investigation or proposed, is helpful for future drug development. However, the description and graphical display (Point 1.) is unclear. For example, line 507-510 could be omitted and more effort should be made to explain the mechanism of the screening for Fra-1 destabilizers, not just putting everything into the figure legends (Fig.4A).
- Section 3.5: Fra-1-based suicide gene therapy is interesting, which, however, is not fully explained. The mechanism of the new drug should be detailed.
Minor points:
Lines19-20, abbreviation "EMT" is not indicated.
Author Response
The review paper “The Fra-1/AP-1 oncoprotein: from the “undruggable” transcription factor to therapeutic targeting” by Casalino et al. describes functional roles of Fra-1 in tumor growth, invasion, metastasis, prognosis and drug resistance. Moreover, the paper summarizes strategies targeting Fra-1 expression, stability, and transcriptional activity, as well as tumor-specific delivery of Fra-1/AP-1-specific drugs. AP-1 transcription factors, including Fra-1, are generally considered “undruggable”. Therefore, the review is interesting and highlights the potential to develop novel agents targeting transcription factors for future cancer treatment.
However, there are several problems that reduce the overall quality of this paper. Here are some suggestions for improvement.
- The quality of the figures is poor and must be improved.
- Fig.1A middle is the same as Figure1 in“AP-1: a double-82 edged sword in tumorigenesis” (https://www.nature.com/articles/nrc1209). However, there is no citation in the figure legend.
- The figure legends are unclear and need rewording.
- There is no indication of panels A-D and A-B in the legend of Figure 1 and Figure 3, respectively.
- The Figures 2, 3 and 5 are difficult to read. Please change the font.
We thank the reviewer for these important comments about the Figures. We have now:
- improved the resolution of the figures inserted in the text (see also the separate jpeg files)
- included the reference (ref. 6) to the Seminal Review by Eferl & Wagner (Nature 2003), in the legend that Fig. 1A ;
- extensively rewritten the Figure legends and indicated the panels A-D and A-B in the legend of Figure 1 and Figure 3;
- increased the size of the fonts of the Figures.
- Sections 2.2, 2.3 and 2.4: graphical display or/and a table to summarize the functional roles and the underlying mechanisms of Fra-1 in tumors will help to make the message clearer.
In response to this precious suggestion, we have now summarized the key target genes and functional roles of Fra-1 in neoplastic cells in a new panel (1B) added to Figure 1.
- Lines 157-164: the sentences could be phrased better.
We have rewritten the whole period (lines…. in the revised manuscript)
- Lines 318-326: The description of Omomyc can be omitted, since it’s not relevant to Fra-1/AP-1.
We have now omitted the description of Omomyc, as requested. In place of the Omomyc, as suggested by the Reviewer #3, we have described the more pertinent c-Jun-derived dominant-negative (TAM67).
- Sections 3.2 and 3.3: The discussion of the strategies targeting Fra-1, which are under investigation or proposed, is helpful for future drug development. However, the description and graphical display (Point 1.) is unclear. For example, line 507-510 could be omitted and more effort should be made to explain the mechanism of the screening for Fra-1 destabilizers, not just putting everything into the figure legends (Fig.4A).
After removing the description of the strategy utilized for screening the inhibitors of c-Myc stability, we have now explained in detail how a similar approach can be adopted for Fra-1 destabilizers (lines 557-567) and modified the related figure (panel 6A).
- Section 3.5: Fra-1-based suicide gene therapy is interesting, which, however, is not fully explained. The mechanism of the new drug should be detailed.
As for the previous point, we have expanded the description of the Fra-1- based suicide gene therapy and detailed the ganciclovir toxicity in the cells in which the HSVtk- Fra-1 is accumulated (lines 759-763).
Minor points:
Lines19-20, abbreviation "EMT" is not indicated.
Specified.
Reviewer 3 Report
Review of Manuscript ID: cancers-1605779
The manuscript by Laura Casalino et al. et al. “The Fra-1/AP-1 oncoprotein: from the “undruggable” transcription factor to therapeutic targeting”, presents a summary of the current knowledge on the FOS-family member and AP-1 transcription factor subunit Fra-1. Fra-1 importance in several aspects of cancer is described and how interfering with its expression, stability, and/or activity could be therapeutically relevant. Strategies aimed at interfering with what is considered an undruggable protein are listed and critically discussed.
This is overall a well written and easy to read review with clear illustrations and most of the relevant papers cited. There are no recent reviews discussing the therapeutic possibilities and challenges of targeting AP-1 and/or its subunits, even though it is well accepted that AP-1 is central to several aspects of tumorigenesis. The present review is therefore timely and comes as a logical continuation to a previously published review by the same authors in 2020 where a small part was touching on the topic (ref 6 in the present review).
Some comments could help improve/clarify the present manuscript:
For non AP-1 experts, the authors should clarify that what is active is Fra-1 containing AP-1 complexes rather than Fra-1 itself since the protein has, in principle, no activity outside heterodimers. Whenever possible the authors should write “Fra-1-containing dimers”, instead of simply Fra-1.
When describing interfering strategies based on modified bzip approaches, such as A-Fos, the authors should probably list early work done with the so called dominant negative c-Jun ie the transactivation domain truncated c-Jun construct (sometimes referred to as TAM67) that was widely used to inhibit AP-1, including in vivo. Some of these efforts are cited in a review by Young and Colburn (PMID: 16784822), that could also be cited.
Finally, the authors should elaborate a bit more on T-5224 : it has been successfully used in preclinical models of cancer (eg in PMID: 26918517) as well as in disease models beyond RA (several references); PMID: 31135379 dealing with the in vivo effects of T-5224 in mice and how it affects the expression of the Fra-1 homologue Fra-2 . Yet, very little is known from the phase II human clinical trials that was mentioned in PMID: 24831826 , and readers would certainly be interested in the outcome of such trial.
Author Response
The manuscript by Laura Casalino et al. et al. “The Fra-1/AP-1 oncoprotein: from the “undruggable” transcription factor to therapeutic targeting”, presents a summary of the current knowledge on the FOS-family member and AP-1 transcription factor subunit Fra-1. Fra-1 importance in several aspects of cancer is described and how interfering with its expression, stability, and/or activity could be therapeutically relevant. Strategies aimed at interfering with what is considered an undruggable protein are listed and critically discussed.
This is overall a well written and easy to read review with clear illustrations and most of the relevant papers cited. There are no recent reviews discussing the therapeutic possibilities and challenges of targeting AP-1 and/or its subunits, even though it is well accepted that AP-1 is central to several aspects of tumorigenesis. The present review is therefore timely and comes as a logical continuation to a previously published review by the same authors in 2020 where a small part was touching on the topic (ref 6 in the present review).
Some comments could help improve/clarify the present manuscript:
For non AP-1 experts, the authors should clarify that what is active is Fra-1 containing AP-1 complexes rather than Fra-1 itself since the protein has, in principle, no activity outside heterodimers. Whenever possible the authors should write “Fra-1-containing dimers”, instead of simply Fra-1.
According to this precious suggestion, especially important for the non AP-1 experts, we have introduced the expression “Fra-1-containing dimers” in various parts of the text, as highlighted.
When describing interfering strategies based on modified bzip approaches, such as A-Fos, the authors should probably list early work done with the so called dominant negative c-Jun ie the transactivation domain truncated c-Jun construct (sometimes referred to as TAM67) that was widely used to inhibit AP-1, including in vivo. Some of these efforts are cited in a review by Young and Colburn (PMID: 16784822), that could also be cited.
This point addresses an important contribution that we have overlooked in our manuscript. We have now described in the text (lines 342-349) and illustrated in the revised figure 3A the TAM67 dominant-negative. Instead of the review by Young and Colburn (PMID: 16784822) mainly focused on Fra-1, we have introduced the reference to the important PNAS publication by the same authors on the transgenic expression of TAM67, in addition to the citation of the original report on TAM67 (refs. 88 and 89).
Finally, the authors should elaborate a bit more on T-5224 : it has been successfully used in preclinical models of cancer (eg in PMID: 26918517) as well as in disease models beyond RA (several references); PMID: 31135379 dealing with the in vivo effects of T-5224 in mice and how it affects the expression of the Fra-1 homologue Fra-2 . Yet, very little is known from the phase II human clinical trials that was mentioned in PMID: 24831826 , and readers would certainly be interested in the outcome of such trial.
We have significantly expanded the section (lines 397-406) on the AP-1 inhibitor T-5224 and introduced new references on preclinical models of non-neoplastic diseases (rheumatoid arthritis, intervertebral disk degeneration and lung fibrosis). Concerning the question on the outcome of the phase II clinical trial mentioned in the review article (ref. 96), on T-5224 for treatment of rheumatoid arthritis, we have been unable to find any information. The trial is not present in the NIH Registry (www.clinicaltrials.gov) and the description available at Japan Registries of Clinical Trials: https://rctportal.niph.go.jp/en/result?q=T-5224&t=chiken&l=10&s=0&c2=desc&o=2does not provide information on the outcome. No information on the website (https://www.fujifilm.com/fftc/en) of the sponsor company (Fujifilm Toyama Chemical Co.) of the clinical trial.
Round 2
Reviewer 2 Report
After revision, the review paper “The Fra-1/AP-1 oncoprotein: from the “undruggable” transcription factor to therapeutic targeting” by Casalino et al. is very well presented. Considering the pivotal role of the AP-1 transcription factors, including Fra-1, in human cancers, and the great progress in targeting these “undruggable” proteins, the review will be very helpful for translational researchers.